# Coronary Artery Disease and Inflammatory Activation Interfere with Peripheral Tissue Electrical Impedance Spectroscopy Characteristics—Initial Report

**DOI:** 10.3390/ijerph20032745

**Published:** 2023-02-03

**Authors:** Tomasz Urbanowicz, Michał Michalak, Ewa Marzec, Anna Komosa, Krzysztof J. Filipiak, Anna Olasińska-Wiśniewska, Anna Witkowska, Michał Rodzki, Andrzej Tykarski, Marek Jemielity

**Affiliations:** 1Cardiac Surgery and Transplantology Department, Poznan University of Medical Sciences, 61-848 Poznan, Poland; 2Department of Computer Science and Statistics, Poznan University of Medical Sciences, 60-806 Poznan, Poland; 3Department of Bionics and Experimental Medical Biology, Poznan University of Medical Sciences, 60-775 Poznan, Poland; 4Department of Hypertensiology, Angiology and Internal Medicine, Poznań University of Medical Sciences, 61-848 Poznan, Poland; 5Institute of Clinical Science, Maria Sklodowska-Curie Medical Academy, 00-136 Warsaw, Poland

**Keywords:** EIS, NLR, coronary artery disease, electrical resistance, electrical capacitance

## Abstract

Background: The electrical properties of cells and tissues in relation to energy exposure have been investigated, presenting their resistance and capacitance characteristics. The dielectric response to radiofrequency fields exhibits polarization heterogeneity under pathological conditions. The aim of the study was to analyze the differences in changes in resistance and capacitance measurements in the range from 1 kHz to 1 MHz, combined with an assessment of the correlation between the results of electrical impedance spectroscopy (EIS) and inflammatory activation. Methods: In the prospective study, EIS was performed on the non-dominant arm in 29 male patients (median (Q1–Q3) age of 69 (65–72)) with complex coronary artery disease and 10 male patients (median (Q1–Q3) age of 66 (62–69)) of the control group. Blood samples were collected for inflammatory index analysis. Results: The logistic regression analysis revealed a negative correlation with inflammatory indexes, including neutrophil to lymphocyte ratio (NLR) in the CAD group in the frequency of 30 kHz (*p* = 0.038, r = −0.317) regarding EIS resistance measurements and a positive correlation in CAD group in the frequency of 10 kHz (*p* = 0.029, r = −0.354) regarding EIS capacitance. Conclusions: The bioelectric characteristics of peripheral tissues measured by resistance and capacitance in EIS differ in patients with coronary artery disease and in the control group. Electrical impedance spectroscopy reveals a statistically significant correlation with inflammatory markers in patients with CAD.

## 1. Introduction

The electrical properties of cells and tissues in relation to energy exposure have been investigated, presenting their permittivity and conductivity [1,2]. Cells and tissues possess resistive and capacitive properties that may be interfered with in pathological conditions [3,4]. Measurements of dielectric polarization of the tissues using a weak electric field application may help in the noninvasive assessment of cell and tissue viability [5]. Volume status can be measured by bioelectrical impedance analysis and helps to estimate body composition as an affordable and accessible method [6]. The bioimpedance method has been used to evaluate hydration status in cardio-renal syndrome [7]. Electric impedance changes may measure not only hydration but also fibrosis [8]. Changes in resistive and capacitive properties of cells and tissues can be estimated by the dielectric spectroscopy method [9]. The dielectric response of cells and tissues to radiofrequency fields has been reported, demonstrating the polarization heterogeneity of vivid structures [10]. The permittivity plays a role in the polarization of the tissue. The external medium or cytoplasm differs in conductivity compared to cell membranes, additionally is involved in the interfacial polarization of the tissue [9] and influences the conductivity of the external medium or cytoplasm in the polarization mechanisms.

The significance of electrical impedance measurements of tissue and cell changes secondary to various pathological stimuli or diseases has been investigated [11]. Microcirculation disturbances secondary to glucose metabolism impairment were suggested by Sieg et al. [12] and in oncology by Das et al. [13]. In a previous analysis, a direct association between pathological stimuli and cellular or tissue response by bioelectric properties was found. The two main measurements obtained in electrical impedance spectroscopy refer to resistance and capacitance. Resistance describes the opposition to the electric flow current of the cells/tissues, and the capacitance presents transmission capability to transfer electric waves [14]. Changes in intercellular resistance to cell migration were reported by Hagiwara et al. [15].

The inflammatory background of coronary artery disease (CAD) may interfere with the electrical characteristics of tissues and cells [16]. Inflammatory activation may be monitored by simple inflammatory markers obtained from whole blood count analysis [17,18]. Although the predictive role of inflammatory markers in advanced coronary artery disease has been proven [19,20], the pathophysiological basis of the relationship between inflammatory activation and coronary atherosclerosis has not been investigated. Brzezinski et al. suggested physicochemical changes measured by dielectric spectroscopy in collagen in comparison to elastin related to peripheral artery disease [21].

The aim of the study was to analyze differences in changes in resistance and capacitance measurements in the frequency range from 1 kHz to 1 MHz between patients with coronary artery disease and healthy subjects. Moreover, the correlation between the results of electrical impedance spectroscopy (EIS) and inflammatory activation assumed on the basis of blood morphology analysis was investigated. The results were verified in the control group to confirm their objectivity.

## 2. Materials and Methods

The prospective study included 29 male patients with median (Q1–Q3) age of 69 (65–72) years with complex coronary artery disease referred for surgical revascularization (CAD group) and 10 male patients with median age of 66 (62–69) representing the healthy control group. The demographical and clinical data are presented in Table 1. The control group consisted of patients with angiographically non-obstructive coronary arteries.

The electrical impedance spectroscopy and whole blood count were performed on admission in CAD patients and the control group. The spectroscopy results were compared and related to the laboratory findings. Thereafter, the results obtained preoperatively and postoperatively (7 days after the operation) were compared.

### 2.1. Whole Blood Count Analysis

Whole blood count analysis included compounds such as white blood counts, neutrophils, monocytes, lymphocytes, platelets measurements, hemoglobin concentration, and hematocrit (Table 2). The inflammatory indexes, including neutrophil to lymphocyte ratio (NLR), monocyte to lymphocyte ratio (MLR), platelet to lymphocyte ratio (PLR), systemic inflammatory response index (SIRI), systemic inflammatory index (SII) and aggregate systemic inflammatory systemic index (AISI) were calculated.

### 2.2. Electrical Impedance Spectroscopy (EIS)

Measurements were performed on the non-dominant forearm. Two electrodes (Ag/AgCl) were placed 10 cm above each other, 5 and 15 cm from the wrist, after skin disinfection, on a dry surface. The analysis was performed on the inner side of the forearm. Only male patients were enrolled in the study to enable the measurements’ reproducibility, as the uncertainty in the electrical measurement of forearm related to gender may relate to natural physiological differences between men and women.

Results for each electrical frequency were obtained after 15 s of application. Resistance (Rp) and capacitance (Cp) measurements were performed using the two-electrode method with an impedance analyzer (HIOKI 3532-50 LCR HiTester (Hioki, Japan)). Measurements in all patients were performed with the alternating voltage of 60 mV (RMS) with the frequency f from 1 kHz to 1 MHz and temperature of 20–25 °C. In order to obtain high-accuracy measurements of the electrical stability of the forearm, the calibration contained in the HIOKI 3532-50 manual was used.

So far, several equivalent circuits describing the measured impedance Z related to the electrical properties of biological materials and the mechanism of electrode polarization (EP) have been proposed [22,23,24,25]. In our study, we consider the equivalent parallel combination of the resistance R_p_ and the capacitance C_p_ of the whole system, which is represented by the series connection of the parallel resistance and capacitance of the forearm with the constant phase element (CPE) related to EP. Hence the measured impedance Z of this circuit is given by:(1)$$Z=\frac{1}{\frac{1}{R_{p}}+j\omega C_{p}}=\frac{d}{{j\omega {\varepsilon^{*}\varepsilon }_{o}S}^{}}$$
where S = 0.78 cm^2^ and d = 10 cm are the Ag/AgCl electrode area and the distance between the electrodes, respectively, ε_o_ is the vacuum permittivity (ε_o_ = 8.854 pF/m), and ω is the angular frequency (ω = 2πf).

The complex permittivity of the entire system is defined as ε* = ε − jσ/ωε_o_, where ε is the relative permittivity of ε and σ is the conductivity (including both DC and AC conductivity). These parameters are the intrinsic electrical properties of the forearm, expressed as intensive quantities, which are independent of the physical size (S and d) of the area of the arm being measured. To interpret the experimental data in Table 3, we transformed the extensive parameters C_p_ and R_p_ in Equation (1) into the equivalent intrinsic parameters ε and σ as follows:(2)$$\varepsilon =\frac{C_{p}d}{{\varepsilon_{o}S}^{}}\;and\;\sigma =\frac{d}{{R_{p}S}^{}}$$

To explain the frequency dependence of the electrical properties of the forearm for the control and CAD groups in the range from 1 kHz to 1 MHz of the electric field, taking into account the EP effect, we used the Cole-Cole model, expressed by the formula:(3)$$\varepsilon^{*}=\varepsilon_{h}+\frac{\Delta \varepsilon }{1+{\left(jf/f_{c}\right)}^{1-\alpha }}+\frac{\sigma_{l}}{{j2\pi f\varepsilon_{o}}^{}}+Af^{-m}$$

σ_l_ is the low-frequency of conductivity for the bulk forearm at a frequency that is also the cut-off frequency below which EP occurs, and Af^−m^ is the EP term of the constants A and m, also known as CPE. The bulk electrical properties of the forearm are represented by: ε_h_ is the high-frequency of relative permittivity, Δε is permittivity decrement characterizing the magnitude of dielectric dispersion, f_c_ is the characteristic frequency, and α denotes the degree of broadening of the dispersion spectra.

According to Table 3, the experimental results of R_p_ and C_p_ and the parameters calculated on the basis of Equations (1)–(3) for the control and CAD groups are presented in the figures as average values.

### 2.3. Exclusion Criteria

Patients requiring combined procedures, including valvular pathology or aortic aneurysm surgery, were excluded from the analysis. Patients with co-existing peripheral artery disease, chronic inflammatory disease, rheumatic disease, and hematological or oncological history were excluded from the study. The electrodes for EIS measurements were placed within 10 cm distance on the forearm. Therefore, females were excluded, and the study group consisted of males only in order to obtain the most accurate results, undisturbed by the anatomical features of the patients.

### 2.4. Statistical Analysis

Analyzed data did not follow normal distribution (Shapiro–Wilk test). The parameters were presented as medians and interquartile ranges (Q1–Q3). The comparison of numerical parameters between CAD before surgery vs. CAD after surgery was performed by Wilcoxon test. For the comparison of CAD groups vs. Control, a Mann–Whitney test was used. The relationship between analyzed parameters was assessed by Spearman’s rank correlation coefficient. The percentages were compared by test for proportions. Statistical analysis was performed with the use of statistical package STATA 17 (StataCorp. 2021. Stata Statistical Software: Release 17. College Station, TX: StataCorp LLC.). All tests were considered significant at *p* < 0.05.

## 3. Results

### 3.1. Laboratory Results

The laboratory tests, including whole blood count analysis, lipid panel, and kidney and liver function tests, were performed on each patient, as presented in Table 2.

### 3.2. EIS Parameters

The results of the EIS between CAD and control groups are presented in Table 3.

The obtained results revealed significant differences between the CAD and control groups regarding electrical resistance and capacitance in the frequency range of 200–500 kHz. The presented results indicate statistically significant differences related to the electrical flow reduction and ions transfer. A graphical comparison of resistance and capacitance between the CAD and control group is presented in Figure 1 and Figure 2.

These spectra show a decrease in the value of both parameters with increasing frequency. However, the spectral shapes of R_p_ above 50 kHz show less variation with frequency compared to the behavior of C_p_. The results in Figure 1 and Figure 2 also show the difference between the control and CAD groups in the values of both of these parameters above 50 kHz. In the 50 kHz–1 MHz range, the control R_p_ values show a slight increase compared to the CAD values. In contrast, in the same frequency range, the C_p_ plot for the control lies below the plot for the CAD.

Figure 3 and Figure 4 present the frequency relationship of the relative permittivity ε and the conductivity σ in the full range of electric field frequencies from 1 kHz to 1 MHz for control and CAD groups. These results for both groups show that as the frequency increases, the ε values decrease and the σ values increase. From the log ε versus log f plots, we estimated the cut-off frequency f_l_ = 50 kHz, below which the EP occurs at a slope m of the straight lines of about 0.8. The range from 50 kHz to 1 MHz is only associated with bulk forearms of the control and CAD groups. For these frequencies, the values of ε and σ for CAD are greater than for controls. This indicates that ion transport (reflected by σ) in the extracellular electrolyte and intracellular cytosol and the ability to accumulate charge (reflected by ε) on the surface of the membranes are more intense in the CAD group.

Figure 5 shows the spectra above 50 kHz for the control and CAD groups, which were obtained using the Cole–Cole model (i.e., Equation (1)) without the term EP. In the presented graphs, the Δε″ (imaginary parts ε*) is defined as (σ − σ_l_)/ωε_0_, taking into account the low conductivity σ_l_ at 50 kHz and the high relative permittivity ε_h_ at 800 kHz. Differences in dielectric behavior between the two groups showed higher CAD spectra compared to the control. This suggests that the number of relaxation active sites and free ions in the CAD is greater than in the control group while maintaining a similar f_c_ between 100 and 200 kHz.

### 3.3. Correlations with Inflammatory Indexes

#### Comparison of the CAD and Control Groups

Logistic regression analysis (Table 4) revealed a negative correlation with inflammatory indexes, including NLR in the CAD group in the frequency of 30 kHz (*p* = 0.038, r = −0.317) regarding EIS resistance measurements, and a positive correlation in the CAD group in the frequency of 10 kHz (*p* = 0.029, r = −0.354) regarding EIS capacitance. The same correlations were not observed in the control group.

## 4. Discussion

Our study revealed two essential observations. First, EIS measurements in patients with complex coronary artery disease revealed significant differences in resistance and capacitance compared to the control group among electrical wave frequencies. The second is the existing correlation between inflammatory markers and EIS results. This is the first study, to our best knowledge, presenting the peripheral tissues’ bioelectrical characteristics related to ischemic heart disease.

Our results provide a new perspective on the pathophysiology of the disease. The bioimpedance properties of peripheral tissues may be influenced by atherosclerotic heart disease. Surgical revascularization may affect the resistance and capacitance of peripheral tissues, as demonstrated in our study. According to our assumptions, coronary artery disease is not the issue of the coronary arteries and muscle by themselves but should be considered as an important factor for peripheral inter- and intracellular bioelectrical properties. Our results suggest some significant differences in peripheral cell resistance secondary to ischemic heart disease. The results may be related to local inflammation, which can easily be assessed by simple hematological indexes.

The cellular components of the tissue determine their electrical properties. Resistance and capacitance are complex variables that depend on the applied electric frequency. Correlation with inflammatory markers in the coronary artery disease group was noticed in the electrical frequency range of 1 kHz to 1 MHz.

The results present the statistically significant correlation between EIS resistance and capacitance at 30 kHz and 10 kHz related to inflammatory activation in patients with complex coronary artery disease. Although the correlation was found in patients with complex coronary disease, it was not more pronounced after surgical revascularization nor in the control group, suggesting the impact of ischemic heart disease on peripheral tissue resistance and capacitance. Therefore, we suggest that the inflammatory milieu related to ischemic conditions strongly influences bioelectrical features. Similarly, differences between healthy and injured tissues were reported in animal experiments. Studies conducted on animals’ lungs by Dean et al., comparing native and damaged tissues, presented similar results [26]. On the other hand, several publications indicate a clear relationship between changes in electrical bioimpedance (EBI) in the wound healing process and inflammation.

Although the EIS results presented a moderate correlation with inflammation, the effect was not found in the control group. These statistically significant results point out a possible reversible relationship between microcirculatory disturbances and coronary artery disease as measured by tissue electrical resistance. The presented relation between inflammatory activation and coronary disease and EIS results was not noticed postoperatively in patients with coronary artery disease. No correlation was observed either in patients without coronary artery atherosclerosis or in patients after surgical revascularization. Thus, we assume that in the conditions of normal cardiac blood supply, bioelectrical activity is not significantly impaired.

This pilot study provides a new perspective on microcirculatory disturbances or the electrical physiology of tissues and cells.

The role of microcirculation dysfunction is strongly associated with the severity of organ dysfunction and mortality [27,28]. The nature and degree of microcirculatory dysfunction are pronounced in patients with inflammatory activation [29]. Electrical frequencies in the 20–30 kHz range related to the microcirculatory environment are characterized by tissue proprieties, including intercellular hydration and protein cumulation. Electrical resistance at relatively low frequencies has been postulated to be related to tissue properties, as reported in E. Marzec et al.’s study [30]. The results of our study indicate the reversibility of microcirculatory disturbances secondary to heart revascularization.

High frequencies of electric stimuli describe the intracellular resistance and capacitance [31,32,33]. The response of peripheral transcriptomic cells to atherosclerotic coronary artery disease is a well-recognized phenomenon [34]. Currently, increasing evidence indicates that such a response has a significant clinical utility in determining CAD-associated events.

The presented results indicate interference between inflammatory markers, including NLR and 500 kHz electric resistance. The obtained results are statistically significant and show possible interaction between inflammatory activation related to atherosclerotic coronary disease and peripheral disturbances of intracellular physiology.

The main limitation of our study is the inability to perform direct EIS measurements performed on heart tissue. However, even if this could be achieved in patients undergoing cardiac surgical revascularization, it would be impossible in the healthy control group. The main strength of our study is the bench-to-bedside analysis.

Since the properties of the cellular structures of biological objects are mainly manifested in the β-dispersion region, our further measurements of the forearm will cover a wide frequency range of up to 10 MHz. In future studies of the electrical properties of the forearms, we will use the four-electrode method to correct the EP effect. The aim of this method will be to shift this polarization towards low frequencies compared to the frequency range below 50 kHz for the EP obtained by the two-electrode method in this study. Due to the masking of α-dispersion by the EP effect, we could not analyze the mechanism of ion conduction and interfacial polarization in the heterogeneous structure of arms made of tissues such as muscle, bone, or fat. Therefore, we expect that the applied four-electrode correction procedure will clearly show the α-dispersion of the electric field during human arm measurements.

## 5. Conclusions

The bioelectric characteristics of peripheral tissues measured by resistance and capacitance in EIS differ in patients with coronary artery disease and the control group. Electrical impedance spectroscopy reveals a statically significant correlation with inflammatory markers in patients with complex coronary artery disease.

## Figures and Tables

**Figure 1 ijerph-20-02745-f001:**
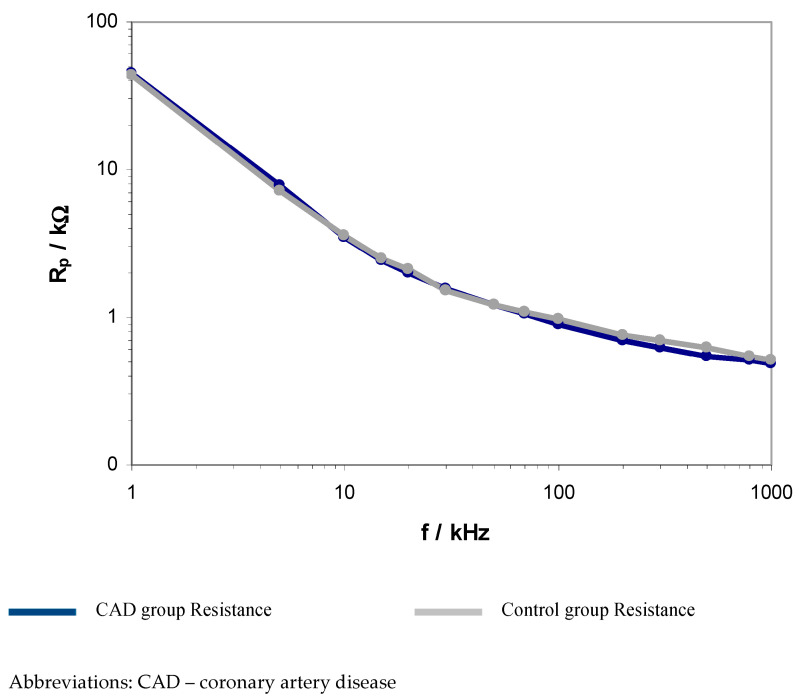
Logarithmic comparison of EIS resistance between CAD and control group within the range from 1 kHz to 1 MHz.

**Figure 2 ijerph-20-02745-f002:**
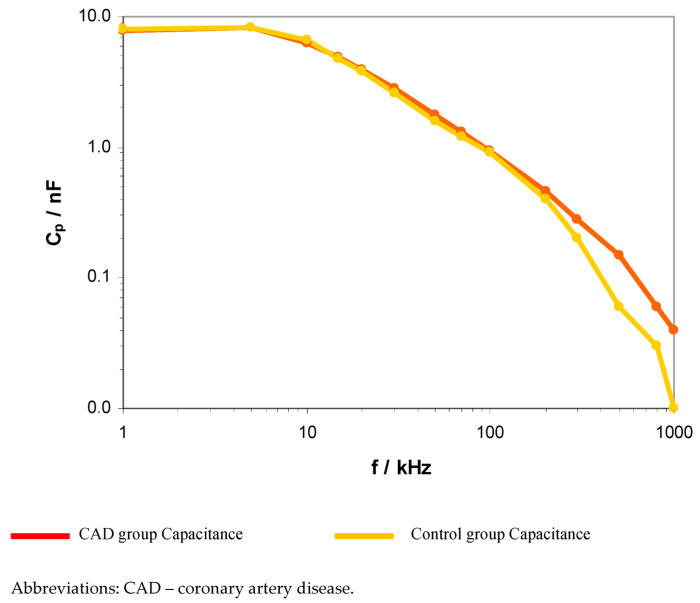
Logarithmic comparison of EIS capacitance between CAD and control group within the range from 1 kHz to 1 MHz.

**Figure 3 ijerph-20-02745-f003:**
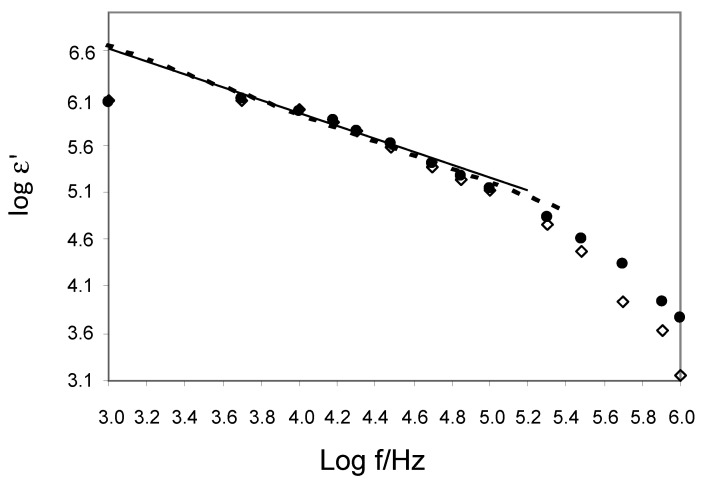
Frequency dependence of the relative permittivity (●—CAD; ◊—control). Solid and dashed lines indicate slope m for CAD and controls.

**Figure 4 ijerph-20-02745-f004:**
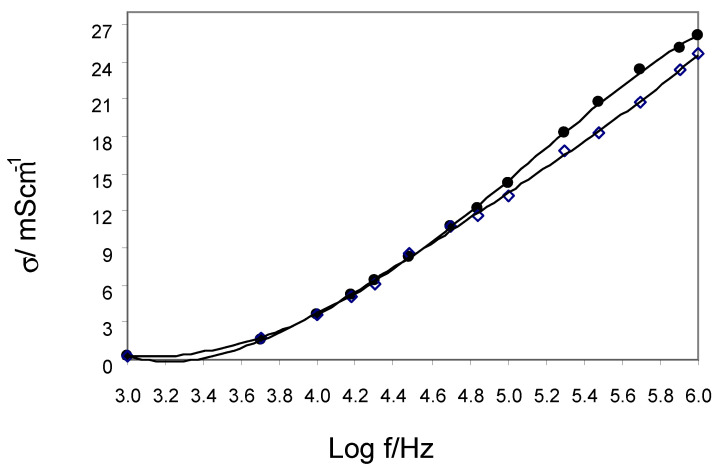
Frequency dependence of the electrical conductivity (●—CAD; ◊—control).

**Figure 5 ijerph-20-02745-f005:**
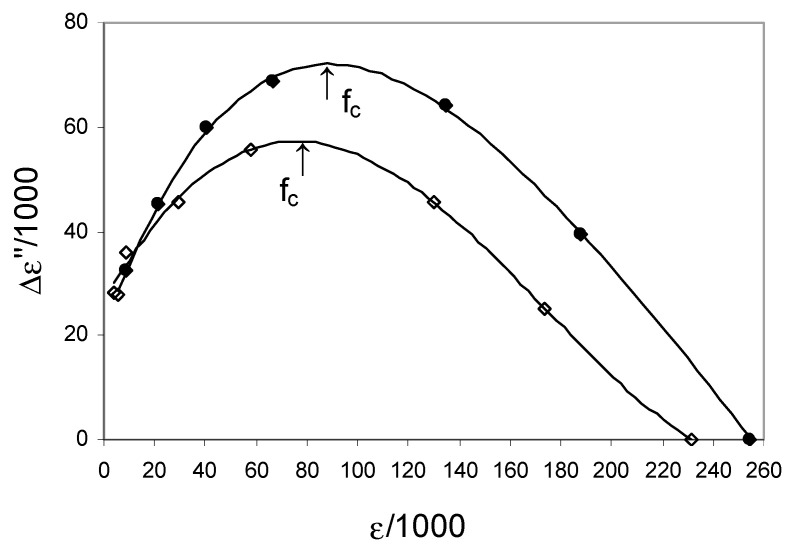
Cole–Cole plots of the experimental data (●—CAD; ◊—control).

**Table 1 ijerph-20-02745-t001:** Demographical and clinical characteristics.

Parameter	CAD Patients *n* = 29	Control Group *n* = 10	*p*
Age (years) (median (Q1–Q3))	69 (65–72)	66 (62–69)	0.786
Sex (male/female) (*n*,%)	29/0	10/0	--
NYHA class (median (Q1–Q3)	2 (1–2)	2 (1–2)	--
Height (cm) (median (Q1–Q3))	175 (170–175)	176 (168–180)	0.698
Weight (kg) (median (Q1–Q3))	86 (79–83)	98 (90–112)	0.134
Co-morbidities:			
Arterial Hypertension (*n*,%)	26 (89%)	10 (100%)	0.567
Diabetes Mellitus (*n*,%)	8 (28%)	5 (50%)	0.356
COPD (*n*,%)	3 (10%)	1 (10%)	0.976
PAD (*n*,%)	0 (0%)	0 (0%)	--
Hypercholesterolemia (*n*,%)	19 (66%)	9 (90%)	0.456
Kidney Failure (*n*,%)	2 (7%)	0	0.187
Atrial Fibrillation (*n*,%)	4 (13%)	0	0.061
Stroke in Medical History (*n*,%)	1 (3%)	0	0.094
Echocardiography			
LV (mm) (median (Q1–Q3))	45 (43–49)	50 (47–52)	0.126
RV (mm) (median (Q1–Q3))	30 (26–33)	32 (31–33)	0.436
LA (mm) (median (Q1–Q3))	37 (33–40)	40 (38–45)	0.438
LVEF (%) (median (Q1–Q3))	60 (60–60)	60 (60–60)	0.976

Abbreviations: CAD—coronary artery disease, COPD—chronic obstructive pulmonary disease, LA—left atrium diameter, LVEF—left ventricle ejection fraction, LV—left ventricle diameter, NYHA—New York Heart Association, PAD—peripheral artery disease, RV—right ventricle diameter.

**Table 2 ijerph-20-02745-t002:** Laboratory test results.

Parameter	CAD before Surgery (A1)	CAD after Surgery (A2)	Control Group (B)	*p* A1 vs. A2	*p* A1 vs. B	*p* A2 vs. B
Laboratory						
WBC (K/uL) (median (Q1–Q3))	7.7 (6.3–9.1)	7.4 (6.1–8.6)	7.7 (6.3–1.1)	0.229	0.035 *	0.098
N (K/uL) (median (Q1–Q3))	5.1 (4.1–6.7)	4.5 (3.9–5.4)	4.3 (3.1–5.2)	0.025 *	0.022 *	0.403
L (K/uL) (median (Q1–Q3))	1.6 (1.2–1.7)	1.7 (1.3–2.0)	1.4 (1.1–1.8)	0.166	0.412	0.139
NLR (median (Q1–Q3))	3.6 (3–4.7)	2.8 (2.4–3.1)	2.9 (2.5–3.5)	0.002 *	0.079	0.541
M (K/uL) (median (Q1–Q3))	0.5 (0.4–0.6)	0.6 (0.5–0.8)	0.4 (0.4–0.5)	0.020 *	0.152	0.004 *
MLR (median (Q1–Q3))	0.4 (0.3–0.4)	0.4 (0.3–0.4)	0.3 (0.3–0.4)	<0.001 *	0.327	0.062
SIRI (median (Q1–Q3))	1.9 (1.4–2.4)	1.8 (1.4–2.1)	1.4 (1.0–1.7)	0.179	0.056	0.094
Eo (K/uL) (median (Q1–Q3))	0.17 (0.08–0.16)	0.35 (0.19–0.35)	0.3 (0.1–0.4)	<0.001 *	0.017 *	0.618
LUC (K/uL) (median (Q1–Q3))	0.12 (0.1–0.18)	0.18 (0.13–0.24)	0.13 (0.12–0.15)	0.003 *	0.923	0.059
Hb (mmol/L) (median (Q1–Q3))	9.2 (8–9.5)	6.9 (6.4–7.2)	9.1 (8.3–9.6)	<0.001 *	0.664	<0.001 *
MCV (fl) (median (Q1–Q3))	92 (90–97)	94 (91–97)	93 (91–96)	0.281	0.746	<0.001 *
RBc (M/uL) (median (Q1–Q3))	4.5 (4.3–4.8)	3.52 (3.31–3.86)	4.7 (4.6–4.9)	<0.001 *	0.234	<0.001 *
Hct (%) (median (Q1–Q3))	43 (40–44)	33 (30–35)	45 (41–47)	<0.001 *	0.061	<0.001 *
MCHC (mmol/L) (median (Q1–Q3))	21.1 (20.4–21.6)	21 (30–35)	20.3 (19.8–20.9)	0.006 *	0.009 *	0.131
RDW (%) (median (Q1–Q3))	13.7 (13.1–14.1)	14 (13.5–19.9)	14 (13.6–14.5)	0.001 *	0.129	0.910
Plt (K/uL) (median (Q1–Q3))	231 (192–279)	221 (214–306)	197 (186–261)	0.006 *	0.499	0.130
SII (median (Q1–Q3))	800 (621–1098)	653 (608–803)	645 (505–849)	0.018 *	0.119	0.585
AISI (median (Q1–Q3))	465 (258–571)	433 (311–544)	287 (261–328)	0.503	0.104	0.031 *
MPV (fl) (median (Q1–Q3))	9 (8.4–9.7)	8.5 (8.1–8.9)	9.1 (8.8–9.9)	0.039 *	0.310	0.016 *
Lipid profile						
Total cholesterol (mmol/L) (median (Q1–Q3))	3.45 (314–3.95)	-	5.1 (4.1–5.7)	-	0.049 *	-
LDL fraction (mmol/L) (median (Q1–Q3))	1.85 (1.49–2.35)	-	2.95 (1.6–3.41)	-	0.274	-
HDL fraction (mmol/L) (median (Q1–Q3))	0.88 (0.84–1.28)	-	1.1 (1.0–1.3)	-	0.378	-
Triglycerides (mmol/L) (median (Q1–Q3))	1.2 (0.82–1.69)	-	1.4 (1.1–1.6)	-	0.421	-
Liver function:						
ALT (U/L) (median (Q1–Q3))	29 (23–35)	31 (20–44)	31 (21–44)	0.768	0.835	0.913
Kidney function:						
Creatinine (umol/L) (median (Q1–Q3))	88 (72–103)	76 (71–82)	55 (79–87)	0.028 *	0.949	0.111
Myocardial markers:		MAX				
Troponin -I (ng/mL) (median (Q1–Q3))	0.012 (0.005–0.02)	1.99 (1.389–2.378)	0.014 (0.006–0.03)	<0.001	0.867	<0.001

Abbreviations: AISI—aggregate index of systemic inflammation, ALT—alanine aminotransferase, CK-MB—creatine kinase myocardial band, Eo—eosinophil count, Hb—hemoglobin, Hct—hematocrit, HDL—high density lipoprotein, L-lymphocytes, LDL—low density lipoprotein, LUC—large unstained cells, M—monocyte count, MCHC—mean corpuscular hemoglobin concentration, MCV—mean corpuscular volume, MLR—monocyte to lymphocyte ratio, MPV—mean platelet volume, N—neutrophil count, NLR—neutrophil to lymphocyte ratio, Plt—platelets, PLR—platelets to lymphocyte ratio, RBC—red blood cells, RDW—red blood cells distribution width, SII—systemic inflammatory index, SIRI—systemic inflammatory response index, WBC—white blood cells. * Significant *p* value.

**Table 3 ijerph-20-02745-t003:** EIS measurements comparison between CAD (A1) and control (B) group.

Frequency f	CAD Group (A1)		Control Group (B)		*p*	
	Rp (kΩ)	Cp (nF)	Rp (kΩ)	Cp (nF)	*p* Rp	*p* Cp
1 kHz	45.3 (35.7–52.7)	7.7 (5.9–8.5)	43.5 (41–49)	8 (6.0–9.1)	0.772	0.515
5 kHz	7.8 (6.13–10.45)	8.3 (7.1–9.9)	7.22 (5.7–8.4)	8.2 (7.6–8.2)	0.106	0.502
10 kHz	3.5 (2.9–3.8)	6.3 (5.8–7)	3.6 (2.7–4.1)	6.6 (6–6.9)	0.306	0.193
15 kHz	2.5 (2.1–2.8)	4.9 (4.5–5.3)	2.5 (1.9–2.7)	4.8 (4.4–4.8)	0.032 *	0.974
20 kHz	2 (1.7–2.4)	3.9 (3.7–4.1)	2.1 (1.6–2.2)	3.8 (3.5–4.1)	0.079	0.987
30 kHz	1.5 (1.3–1.6)	2.8 (2.6–3.0)	1.4 (1.3–1.6)	2.6 (2.4–2.7)	0.880	0.129
50 kHz	1.2 (1–1.3)	1.7 (1.6–1.9)	1.2 (1–1.2)	1.6 (1.3–1.7)	0.065	0.879
70 kHz	1 (0.9–1.1)	1.3 (1.2–1.4)	1.1 (1–1.2)	1 (0.1–1.1)	0.943	0.678
100 kHz	0.9 (0.8–1.0)	0.9 (0.9–1.0)	0.9 (0.9–1.0)	0.9 (0.9–1.1)	0.066	0.508
200 kHz	0.7 (0.6–0.7)	0.5 (0.4–0.5)	0.8 (0.8–0.9)	0.4 (0.2–0.7)	0.035 *	0.001 *
300 kHz	0.6 (0.6–0.7)	0.3 (0.3–0.3)	0.7 (0.7–0.7)	0.2 (0.2–0.2)	0.045 *	0.008 *
500 kHz	0.5 (0.4–0.6)	0.1 (0.05–0.2)	0.6 (0.5–0.6)	0.06 (0.06–0.07)	0.047 *	0.003 *
800 kHZ	0.51 (0.48–0.53)	0.06 (0.05–0.08)	0.55 (0.41–0.63)	0.03 (0.02–0.04)	0.224	0.004 *
1 MHZ	0.49 (0.45–0.52)	0.04 (0.03–0.05)	0.51 (0.48–0.52)	0.01 (0.01–0.01)	0.628	0.001 *

Abbreviations: Rp—resistance, Cp—capacitance, kΩ—kiloohms, nf—nanofarads. * Significant *p* value.

**Table 4 ijerph-20-02745-t004:** Correlation between EIS measurements and inflammatory markers.

Frequency f	Inflammatory Index	Rp/Cp	CAD Group (A1)		Control Group (B)	
			Spearman’s Rho	*p*	Spearman’s Rho	*p*
1 kHz	NLR	Rp	0.045	0.815	−0.571	0.084
5 kHz	NLR	Rp	−0.254	0.749	−0.628	0.052
10 kHz	NLR	Rp	−0.405	0.101	−0.246	0.493
15 kHz	NLR	Rp	−0.329	0.059	−0.238	0.724
20 kHz	NLR	Rp	0.049	0.587	−0.337	0.136
30 kHz	NLR	Rp	−0.317	0.038 *	−0.628	0.052
50 kHz	NLR	Rp	−0.252	0.069	−0.261	0.466
70 kHz	NLR	Rp	−0.235	0.529	−0.409	0.241
100 kHz	NLR	Rp	−0.368	0.060	−0.523	0.121
200 kHz	NLR	Rp	0.007	0.465	−0.415	0.233
300 kHz	NLR	Rp	0.009	0.538	−0.175	0.629
500 kHz	NLR	Rp	−0.161	0.047 *	−0.525	0.119
800 kHz	NLR	Rp	0.270	0.611	−0.174	0.631
1 MHz	NLR	Rp	0.028	0.067	−0.174	0.630
1 kHz	NLR	Cp	−0.126	0.514	−0.281	0.433
5 kHz	NLR	Cp	0.311	0.343	−0.246	0.493
10 kHz	NLR	Cp	0.354	0.029 *	−0.534	0.802
15 kHz	NLR	Cp	0.147	0.082	−0.238	0.724
20 kHz	NLR	Cp	0.388	0.856	−0.506	0.136
30 kHz	NLR	Cp	−0.317	0.094	−0.628	0.052
50 kHz	NLR	Cp	0.167	0.187	−0.261	0.466
70 kHz	NLR	Cp	0.381	0.354	−0.409	0.241
100 kHz	NLR	Cp	0.197	0.049 *	−0.523	0.121
200 kHz	NLR	Cp	0.166	0.978	0.233	0.031
300 kHz	NLR	Cp	0.09	0.972	−0.175	0.629
500 kHz	NLR	Cp	−0.162	0.402	−0.525	0.119
800 kHz	NLR	Cp	0.345	0.312	0.323	0.631
1 MHz	NLR	Cp	−0.166	0.883	−0.640	0.631

Abbreviations: CAD—coronary artery disease, Cp—capacitance, NLR—neutrophils to lymphocyte ratio, Rp—resistance. * Significant *p* value.

## Data Availability

Data will be available for 3 years following publication via contacting the corresponding author (tomasz.urbanowicz@skpp.edu.pl) after reasonable requirements.

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
