# Peer review of "Coronary Artery Disease and Inflammatory Activation Interfere with Peripheral Tissue Electrical Impedance Spectroscopy Characteristics—Initial Report"

_ijerph, 2023, doi:10.3390/ijerph20032745_

Round 1

Reviewer 1 Report (Previous Reviewer 1)

I think that, in its present form, the article can be published in your Special Issue, but, in my opinion, there are still, though, two points that the authors should properly address:

 1. Although, they changed the sentence in lines 51-52, I still think that the authors have a confusion there, and they are probably mixing up conductivity with permittivity. Also, when they mention "external medium or cytoplasm" ¿Do they mean either of them or both? Permittivity is actually what plays a role in "polarization of the tissue".

 2. For me, it remains unclear what the position of the electrodes has to do with excluding women. In other words, ¿What would be the explanation for that from the physical point of view?

Best wishes.

Author Response

Dear Reviewer,

I would like to thank you for your valuable comments and suggestions.

Please, find enclosed answers:

Detailed Responses to Reviewer # 1:

The Reviewer’s comment:

  1. Although, they changed the sentence in lines 51-52, I still think that the authors have a confusion there, and they are probably mixing up conductivity with permittivity. Also, when they mention "external medium or cytoplasm" ¿Do they mean either of them or both? Permittivity is actually what plays a role in "polarization of the tissue".
    The Authors’ answer:

The Reviewer is right that: "Permittivity is actually what plays a role in "polarization of the tissue". However, we also assume the influence of the conductivity of the external medium or cytoplasm on the polarization mechanism.

The Reviewer’s comment:

  1. For me, it remains unclear what the position of the electrodes has to do with excluding women. In other words, ¿What would be the explanation for that from the physical point of view?

The Authors’ answer:

The total uncertainty in the electrical measurement of forearm it may also be due to natural physiological differences between men and women. This sex selection procedure is beneficial for measurement reproducibility.

Kind regards

Tom Urbanowicz

Reviewer 2 Report (Previous Reviewer 2)

This is a second-round review  (R2) for the resubmitted manuscript.

The Reviewer’s comment (R1):

The authors also claim that the effect of inflammation and surgical revascularization was clearly visible only in patients with complex coronary artery disease (lines 182 - 186). This is unexpected as several publications report a clear relationship between changes in EBI in the process of wound healing and inflammation.

The Authors’ answer:

Following the reviewer's comment, the revised version addresses the following places in the manuscript:  in Discussion, lines 266-268.

Reviewers comment (R2): A reviewer's note is included but without references and clarification.

The Reviewer’s comment (R1):

1. The properties of the cellular structures of biological objects are manifested mainly in the β-dispersion area, which typically covers the frequency range of up to 10 MHz. Limiting the frequency band to 1 MHz (as in the presented work) introduces significant model parameterization errors.

The Authors’ answer:

Following the reviewer's comment, the revised version addresses the following places in the manuscript: in Discussion, lines 302-312

The Reviewer’s comment (R2):

Authors agree that measurements up to 10 MHz must be made in the future, but the current results remain questionable since it has not been done.  It is also claimed that the 4-electrode method will be used to eliminate the effect of electrode polarization, which is incorrect because 1) non-polarizable electrodes are employed; 2) it remains unclear what is the meaning of the "elimination of electrode polarization effect".

The Reviewer’s comments (R1):

2. The authors use a two-element (Rp, Cp) electrical model of the object. However, almost 100 years ago, it was established that at least three elements (Fricke-Morse models A and B) are needed to model cellular fluids and tissues. In addition, the so-called constant-phase element (CPE) must be used instead of capacitance, which has two parameters. It has been shown that a simplified model (e.g., replacing CPE with a capacitance) leads to significant differences in model parameters. The two-element model corresponds the least to the actual situation.

4. There are no detailed data on the used electrodes, and the impedance part of the electrodes is not specified in the measurement results. However, since the resistance of gel ECG  Ag/AgCl electrodes may be in the order of kilo-ohm, this may significantly impact total  results. What is the typical mean value and variation of the impedance due to the used electrodes?

The Authors’ answer:

Following the reviewer's comment, the revised version addresses the following places in the manuscript: in Materials and methods, lines 107-145, in Results, lines 187-225

The Reviewers comment (R2):

The model used is not clearly described and illustrated. However,  based on the description, it can be concluded that it is unsuitable for the given situation. The CPE is attributed to missing polarized electrodes, and the presented effective capacitance versus permittivity relationships are inadequate. There is no answer to question 4.

The Reviewer’s comment (R1)

3. EIS measurements were performed on the wrist at variable electrode distances of 5-15 cm (lines 98-100). No attention is paid to the fact that the measurement results also depend on distance. What is the typical impedance variation from a distance change from 5 to 15 cm?

The Authors’ answer: 

Following the reviewer's comment, the revised version addresses the following places in the manuscript:  in Materials and methods, lines 99-101.

The Reviewers comment (R2): Ok.

The Reviewer’s comment (R1):

5. The authors do not consider and analyze the effect of the surrounding environment (stray impedances of the body, connection cables and the measuring instrument) on the EIS measurement results, which typically manifests considerably at frequencies above 100 kHz.

The Authors’ answer:

Following the reviewer's comment, the revised version addresses the following places in the manuscript:  in Materials and methods, lines 107-108.

The Reviewers comment (R2):

The standard procedure for calibration typically covers only the situation where the object is located near the connection sockets (or adapter) of the LCR measuring instrument. Considering the effect of longer connecting cables and larger objects requires additional measures. The reader of the article does not need to study the manuals, and the authors must provide a sufficient description of the specific situation (a drawing or photo of the arrangement of the measuring instrument, object and connecting cables, together with a description of the correction of the effect of parasitic impedances.

The Reviewers conclusion (R2)

The authors have made efforts to improve the content of the manuscript. Unfortunately, they did not prove sufficient. There is no good understanding of bioimpedance modelling and measurement principles.

Not all inconsistencies are described in detail by the reviewer since the explanations would have been too long.

Author Response

Dear Reveiwer,

I would like to thank you for your comments.

Please, find enclosed answers to your questions.

The Authors’ answer:

We disagree with the reviewer's opinion that: "The authors have made efforts to improve the content of the manuscript. Unfortunately, they did not prove sufficient. There is no good understanding of bioimpedance modelling and measurement principles.
Not all inconsistencies are described in detail by the reviewer since the explanations would have been too long."

We consider the forearm model used by us to be correct and fully justified to assign the EP electrode polarization to CPE.

The following statements by the Reviewer about the Ag/AgCl electrodes we use are incorrect:

“…….1) non-polarizable electrodes are employed; 2) it remains unclear what is the meaning of the "elimination of electrode polarization effect".”

After all, it has long been known, based on human physico-chemical and biological research, that an electric charge accumulates at the interface of phases with different conductivity, and thus a polarization mechanism occurs. And this is also the case in our research, because the skin of the forearm and the electrodes differ in dielectric properties, hence the response of the electrodes must be EP. According to current scientific publications, electrical studies of the EP-human tissue system are still being analyzed, which is why we proposed a four-electrode method, also used in clinical trials, as a continuation of our research.

We kindly inform you that all answers included in the manuscript in response to the reviewer's comment (R1) no longer require proofreading.

Kind regards

Tom Urbanowicz

This manuscript is a resubmission of an earlier submission. The following is a list of the peer review reports and author responses from that submission.

Round 1

Reviewer 1 Report

MINOR ASPECTS

Abstract, line 28: NLR in CAD are mentioned for the first time but their meanings are not presented.

Introduction, lines 42 & 43: I do not understand what the authors mean by "The volume overload"

Introduction lines 50 & 51: In my opinioin, this asseretion has no meaning "The external medium or cytoplasm are less conductive than the cell membranes and presented discrepancies play a central role in interfacial polarization".

Exclusion criteria, lines 100 & 11: I do not get what the sentece "As the electrodes for EIS measurements were placed within 10 cm distance on the forearm." has to do with exclusion criteria.

MAJOR ASPECTS

In my opinion, this study is not scientific sound, as the conclusions presented in it can not be realated to what they claim. For instance, ¿How can they state that the electrical readings are, say, solely "...related to cardiac blood suply" (lines 197 & 198? My impression is that the study represents a kind of shotgun approach, where they have a lot of data and find some differences and correlations that ought well to be spurious.

On the other side, the impedance measurements carried out in the study may not be adequate, firstly, by taking two electrode measurements, and, secondly, because the results can be affected by multiple variables not taken into account as the volume of the segments and their composition in terms of, for instance, fat and muscle.

I also considere that, when using EBIS measurements, it is more adequate to work in terms of impeditivity, rather than resistance and reactance. It is also more desirable to use calculated parameters rather than raw data. 

Author Response

Dear Reviewer, we corrected the manuscript according to your valuable comments.

The Reviewer’s comment:

MINOR ASPECTS

Abstract, line 28: NLR in CAD are mentioned for the first time but their meanings are not presented.

The Authors’ answer:

Following the reviewer's comment, we explained the abbreviation.

The Reviewer’s comment:

MINOR ASPECTS

Introduction, lines 42 & 43: I do not understand what the authors mean by "The volume overload"

The Authors’ answer:

We referred to the paper by van der Sande et al. who assessed overhydration - fluid overload – in patients with chronic kidney disease treated with different modes of dialysis. Following the reviewer's comment, we changed the “overload” to “status” which, as we hope, would better clarify the sentence meaning.

The Reviewer’s comment:

MINOR ASPECTS

Introduction lines 50 & 51: In my opinioin, this asseretion has no meaning "The external medium or cytoplasm are less conductive than the cell membranes and presented discrepancies play a central role in interfacial polarization".

The Authors’ answer:

Following the reviewer's comment, the revised version addresses the following places in the manuscript:

in Introduction, lines 51-52.

The Reviewer’s comment:

MINOR ASPECTS

Exclusion criteria, lines 100 & 11: I do not get what the sentece "As the electrodes for EIS measurements were placed within 10 cm distance on the forearm." has to do with exclusion criteria.

The Authors’ answer:

Following the reviewer's comment, we clarified the sentences – lines 151-154.

The Reviewer’s comment:

MAJOR ASPECTS

In my opinion, this study is not scientific sound, as the conclusions presented in it can not be realated to what they claim. For instance, ¿How can they state that the electrical readings are, say, solely "...related to cardiac blood suply" (lines 197 & 198? My impression is that the study represents a kind of shotgun approach, where they have a lot of data and find some differences and correlations that ought well to be spurious.

I also considere that, when using EBIS measurements, it is more adequate to work in terms of impeditivity, rather than resistance and reactance. It is also more desirable to use calculated parameters rather than raw data. 

The Authors’ answer:

Following the reviewer's comment, the revised version addresses the following places in the manuscript:

in Materials and methods, lines 107-145,

in Results, lines 187-225.

The Reviewer’s comment:

MAJOR ASPECTS

On the other side, the impedance measurements carried out in the study may not be adequate, firstly, by taking two electrode measurements, and, secondly, because the results can be affected by multiple variables not taken into account as the volume of the segments and their composition in terms of, for instance, fat and muscle.

The Authors’ answer:

Following the reviewer's comment, the revised version addresses the following places in the manuscript:

in Discussion, lines 302-312.

Kind regards

Tomasz Urbanowicz

on behalf of all co-authors 

Reviewer 2 Report

The authors have done extensive work in conducting experiments and statistical analysis. They claim to have identified the association of cardiovascular disease and inflammation with electrical impedance characteristics, which is the first study of its kind related to ischemic heart disease.

However, several works have already investigated the relationship between tissue inflammation and electrical bio-impedance (EBI). Moreover, as the measurements are carried out on a wrist, their EBI relationship with heart muscle properties is very indirect (which the authors partly admit, lines 218-221). The authors also claim that the effect of inflammation and surgical revascularization was clearly visible only in patients with complex coronary artery disease (lines 182 - 186). This is unexpected as several publications report a clear relationship between changes in EBI in the process of wound healing and inflammation. The results of statistical analysis and conclusions are intriguing but not sufficiently convincing because of ignoring several basic quality concepts of EBI modelling and measurement, as described in more detail below.

1. The properties of the cellular structures of biological objects are manifested mainly in the β-dispersion area, which typically covers the frequency range of up to 10 MHz. Limiting the frequency band to 1 MHz (as in the presented work) introduces significant model parameterization errors.

2. The authors use a two-element (Rp, Cp) electrical model of the object. However, almost 100 years ago, it was established that at least three elements (Fricke-Morse models A and B) are needed to model cellular fluids and tissues. In addition, the so-called constant-phase element (CPE) must be used instead of capacitance, which has two parameters. It has been shown that a simplified model (e.g., replacing CPE with a capacitance) leads to significant differences in model parameters. The two-element model corresponds the least to the actual situation.

3. EIS measurements were performed on the wrist at variable electrode distances of 5-15 cm (lines 98-100). No attention is paid to the fact that the measurement results also depend on distance. What is the typical impedance variation from a distance change from 5 to 15 cm?

4. There are no detailed data on the used electrodes, and the impedance part of the electrodes is not specified in the measurement results. However, since the resistance of gel ECG Ag/AgCl electrodes may be in the order of kilo-ohm, this may significantly impact total results. What is the typical mean value and variation of the impedance due to the used electrodes?

5. The authors do not consider and analyze the effect of the surrounding environment (stray impedances of the body, connection cables and the measuring instrument) on the EIS measurement results, which typically manifests considerably at frequencies above 100 kHz.

Conclusion: Since the used model does not correspond to the physical structure of the object and the accuracy of EIS measurements is not analyzed, the reliability of the presented results is questionable. It is very doubtful whether using such a simplified model and careless interpretation of measurement results can adequately reflect the declared correlations with the object's properties. To prove the opposite, it is necessary to describe the measurements' conditions fully and analyze the variation of the results due to measurement errors and the use of the simplified model.

Author Response

Dear Reviewer,

we thank you for the insightful comments. We corrected the manuscript according to them.

The Reviewer’s comment:

The authors also claim that the effect of inflammation and surgical revascularization was clearly visible only in patients with complex coronary artery disease (lines 182 - 186). This is unexpected as several publications report a clear relationship between changes in EBI in the process of wound healing and inflammation.

The Authors’ answer:

Following the reviewer's comment, the revised version addresses the following places in the manuscript:

in Discussion, lines 266-268.

The Reviewer’s comment:

  1. The properties of the cellular structures of biological objects are manifested mainly in the β-dispersion area, which typically covers the frequency range of up to 10 MHz. Limiting the frequency band to 1 MHz (as in the presented work) introduces significant model parameterization errors.

The Authors’ answer:

Following the reviewer's comment, the revised version addresses the following places in the manuscript:

in Discussion, lines 302-312.

The Reviewer’s comment:

  1. The authors use a two-element (Rp, Cp) electrical model of the object. However, almost 100 years ago, it was established that at least three elements (Fricke-Morse models A and B) are needed to model cellular fluids and tissues. In addition, the so-called constant-phase element (CPE) must be used instead of capacitance, which has two parameters. It has been shown that a simplified model (e.g., replacing CPE with a capacitance) leads to significant differences in model parameters. The two-element model corresponds the least to the actual situation.
  2. There are no detailed data on the used electrodes, and the impedance part of the electrodes is not specified in the measurement results. However, since the resistance of gel ECG Ag/AgCl electrodes may be in the order of kilo-ohm, this may significantly impact total results. What is the typical mean value and variation of the impedance due to the used electrodes?

The Authors’ answer:

Following the reviewer's comment, the revised version addresses the following places in the manuscript:

in Materials and methods, lines 107-145,

in Results, lines 187-225.

The Reviewer’s comment:

  1. EIS measurements were performed on the wrist at variable electrode distances of 5-15 cm (lines 98-100). No attention is paid to the fact that the measurement results also depend on distance. What is the typical impedance variation from a distance change from 5 to 15 cm?

The Authors’ answer:

Following the reviewer's comment, the revised version addresses the following places in the manuscript:

in Materials and methods, lines 99-101.

The Reviewer’s comment:

  1. The authors do not consider and analyze the effect of the surrounding environment (stray impedances of the body, connection cables and the measuring instrument) on the EIS measurement results, which typically manifests considerably at frequencies above 100 kHz.

The Authors’ answer:

Following the reviewer's comment, the revised version addresses the following places in the manuscript:

in Materials and methods, lines 107-108.

Kind regards

Tomasz Urbanowicz

on behalf of all co-authors

Reviewer 3 Report

This is a very interesting area of research.

However, in terms of the quality of the scientific work, some details are still missing:

1. A mechanistic description, e.g. modeling of EIS is missing, (i.e. using equivalent R-C circuit). See for example: Sensors 2016, 16, 1900; doi:10.3390/s16111900

2. Which EIS sensor was used? Example: Microelectrodes, etc.

3. Please describe the measurement setup.    
For your inspiration, I may recommend some literature for your scientific work and to improve this manuscript:

> https://www.ncbi.nlm.nih.gov/pmc/articles/PMC8512860/
> Biomed J Sci & Tech Res  2021 35 (2) , 27548, DOI: 10.26717/BJSTR.2021.35.005682

I look forward to receiving a revised version of your manuscript. 

Author Response

Dear Reviewer, we corrected the manuscript according to your insightful comments.

The Reviewer’s comment:

  1. A mechanistic description, e.g. modeling of EIS is missing, (i.e. using equivalent R-C circuit). See for example: Sensors 2016, 16, 1900; doi:10.3390/s16111900
    3. Please describe the measurement setup.    
    For your inspiration, I may recommend some literature for your scientific work and to improve this manuscript:
    > https://www.ncbi.nlm.nih.gov/pmc/articles/PMC8512860/
    > Biomed J Sci & Tech Res  2021 35 (2) , 27548, DOI: 10.26717/BJSTR.2021.35.005682
    The Authors’ answer:

Following the reviewer's comment, the revised version addresses the following places in the manuscript:

in Materials and methods, lines 109-115.

The Reviewer’s comment:

  1. Which EIS sensor was used? Example: Microelectrodes, etc.
    The Authors’ answer:

Following the reviewer's comment, the revised version addresses the following places in the manuscript:

in Materials and methods, lines 99-101.

Kind regards

Tomasz Urbanowicz

On behalf of all co-authors